# Phytohormones as Regulators of Mitochondrial Gene Expression in *Arabidopsis thaliana*

**DOI:** 10.3390/ijms242316924

**Published:** 2023-11-29

**Authors:** Ivan A. Bychkov, Elena S. Pojidaeva, Anastasia S. Doroshenko, Vladimir A. Khripach, Natalia V. Kudryakova, Victor V. Kusnetsov

**Affiliations:** 1Timiryazev Institute of Plant Physiology, Russian Academy of Sciences, 127276 Moscow, Russia; ivan.a.b@mail.ru (I.A.B.); alenapoj@mail.ru (E.S.P.); anastasiya04101993@gmail.com (A.S.D.); 2Institute of Bioorganic Chemistry, National Academy of Sciences of Belarus, 220072 Minsk, Belarus; iboch_steroids@yahoo.com

**Keywords:** *Arabidopsis* *thaliana*, gene expression, mitochondria, phytohormones, RPOTmp

## Abstract

The coordination of activities between nuclei and organelles in plant cells involves information exchange, in which phytohormones may play essential roles. Therefore, the dissection of the mechanisms of hormone-related integration between phytohormones and mitochondria is an important and challenging task. Here, we found that inputs from multiple hormones may cause changes in the transcript accumulation of mitochondrial-encoded genes and nuclear genes encoding mitochondrial (mt) proteins. In particular, treatments with exogenous hormones induced changes in the *GUS* expression in the reporter line possessing a 5′-deletion fragment of the *RPOTmp* promoter. These changes corresponded in part to the up- or downregulation of *RPOTmp* in wild-type plants, which affects the transcription of mt-encoded genes, implying that the promoter fragment of the *RPOTmp* gene is functionally involved in the responses to IAA (indole-3-acetic acid), ACC (1-aminocyclopropane-1-carboxylic acid), and ABA (abscisic acid). Hormone-dependent modulations in the expression of mt-encoded genes can also be mediated through mitochondrial transcription termination factors 15, 17, and 18 of the mTERF family and genes for tetratricopeptide repeat proteins that are coexpressed with *mTERF* genes, in addition to *SWIB5* encoding a mitochondrial SWI/SNF (nucleosome remodeling) complex B protein. These genes specifically respond to hormone treatment, displaying both negative and positive regulation in a context-dependent manner. According to bioinformatic resources, their promoter region possesses putative *cis*-acting elements involved in responses to phytohormones. Alternatively, the hormone-related transcriptional activity of these genes may be modulated indirectly, which is especially relevant for brassinosteroids (BS). In general, the results of this study indicate that hormones are essential mediators that are able to cause alterations in the transcript accumulation of mt-related nuclear genes, which, in turn, trigger the expression of mt genes.

## 1. Introduction

The endosymbiotic organelles of eukaryotic cells, plastids, and mitochondria are tightly integrated into cellular signaling networks as inseparable parts of the plant cell that are needed for photosynthesis and ATP production [1]. The coordinated expression of the organellar and nuclear genomes is achieved through a variety of signals, among which phytohormones make essential contributions. The effects of various hormones on the expression of chloroplast genes are well documented. Exogenously supplied methyl jasmonate (MJ), IAA, ABA, and gibberellic acid (GA) repressed the transcription and transcript accumulation of plastid genes, while cytokinins (CKs) counteracted their action [2]. However, the molecular mechanisms underlying the transduction of hormonal signals to plastids are still poorly understood. Despite the fact that chloroplasts are sites for the production of a number of hormones or their precursors (CK, ABA, SA, IAA, jasmonic acid, and ethylene) [3], the plastid genome does not retain the genes responsible for the perception and transduction of hormonal signals. Therefore, all stages in the hormone-dependent expression of plastid genes are determined primarily via anterograde signals coming from the nucleus.

Research into the interactions between phytohormones and mitochondria has mainly focused on stress-related aspects [4], and despite some progress, the mechanisms of such interactions are far from being fully understood. The Arabidopsis mitochondrial genome contains 58 genes encoding tRNAs, rRNAs, ribosomal proteins, and subunits of the respiratory chain, in addition to 42 noncoding genes [1]. They are transcribed by two nuclear-encoded page-type RNA polymerases: RPOTm, which is exclusively targeted to the mitochondria, and RPOTmp, which is found only in dicots and is bidirected to the mitochondria and chloroplasts. RPOTm is vital for plant development, as its disruption was found to be lethal [5]. RPOTmp is needed to optimally transcribe a subset of mitochondrial genes, including those for respiratory chain complexes I and IV [5]. A ack of this enzyme causes mitochondrial dysfunction, resulting in a strongly reduced mitochondrial respiratory chain and a compensatory upregulation of alternative oxidase (AOX)-dependent respiration. In addition to functions in the mitochondria, RPOTmp was shown to transcribe the *rrn* operon from the *PC* promoter in plastids during seed imbibition [6].

The effects of hormones on organellar gene expression (OGE) can at least partially be transduced through the genes for transcription machinery. In our previous works, we showed that CK-induced changes in the expression of genes encoding chloroplast RNA polymerases and polymerase-associated proteins (PAPs) resulted in the modulated expression of chloroplast genes, suggesting a role for the transcription apparatus in their hormone-dependent regulation [7,8]. However, the exact way in which components of the transcription apparatus induce or suppress the transcription of plastid genes is not known. Hormone-related shifts can also be regulated by the organellar-specific import of transcription factors, providing direct binding to transcription initiation sites and conferring promoter specificity in organellar transcription. To date, a convincing example has been presented for the ABA-dependent transcription of the chloroplast *psbD* gene from the blue-light-responsive promoter (*BLRP*) via the ABA-dependent stimulation of SIG5–PEP-dependent transcription [9].

Among the OGE regulators affected by hormones (bar.utoronto.ca/efp/cgi-bin/efpWeb.cgi (accessed on 2 October 2023)) are proteins of the mitochondrial transcription termination factor family (mTERF) and tetra- and pentatricopeptide repeat proteins (TPRs and PPRs), which are coexpressed with the *mTERF* genes of the mitochondrial cluster [10]. Another potential candidate is a mitochondrial SWI/SNF (nucleosome remodeling) complex B protein, SWIB5, which is capable of associating with mitochondrial DNA (mtDNA) and influencing the mtDNA architecture in *Arabidopsis thaliana* [11]. Whether these regulatory proteins are engaged in the transduction of hormonal signals to mitochondria and whether they direct the expression of mt genes remain to be seen.

The accumulating data suggest that in addition to anterograde signaling, the coordinated expression of organellar and nuclear genomes can be achieved through mechanisms of hormone-dependent retrograde signaling. In particular, Wang and Auwerx [4] established that proteotoxic stress in mitochondria caused by the accumulation of unassembled or unfolded proteins culminates in a systemic hormone response that is mainly reliant on ethylene signaling but also involves auxin and jasmonate. Blocking ethylene signaling partially suppressed mitochondria-to-nucleus signaling, most likely independently of the transcription factor ANAC017, a key regulator of mitochondrial proteotoxic stress responses in plants [12]. Contrary to these data, Meridino et al. [1] showed that a mutation in *RPOTmp* that caused defects similar to the triple response in the dark needed ANAC017. These contradictory results are explained as a result of a weak positive feedback loop linking ethylene and the ANAC017-dependent regulation of mitochondrial retrograde signaling. Anyway, these data indicate that hormones are integral factors in regulating the coordinated expression of the organellar and nuclear genomes. However, there is only limited information regarding their exact functions in this process.

In this work, we found that inputs from multiple hormones may cause context-dependent alterations in the transcript accumulation of genes for mt RNA polymerases as well as MTERF and SWIB family genes, which play roles in modulating the expression changes of mt genes.

## 2. Results

### 2.1. Plant Hormones Trigger Alterations in Mitochondrial Transcript Abundance

To monitor the signals involved in plant responses to exogeneous hormones, we profiled transcripts of some typical markers in 10-day-old wild-type seedlings exposed to hormone treatment for 3 h. The concentrations of hormones and the duration of treatment were selected in the preliminary experiments. The results indicate the upregulation of the target gene transcripts specific for CK (*ARR5*), IAA (*IAA19*), ACC (*ERF1*), ABA (*RD29*), and SA (*PR1*), or, conversely, the downregulation of brassinosteroids (*DWF4*) and GA (*GA_3_*), indicating the effectiveness of the treatment (Appendix A).

To address the hormone-induced changes of the mt-encoded genes, we performed a qRT–PCR analysis. Differentially expressed genes with a ratio of transcript change of 1.5 in at least two tests were classified as regulated [1,13]. 

Twelve genes were selected for the analysis, which represent the main functional groups of the mitochondrial genome. The analysis included genes for the subunits of complexes I (*nad3* and *nad6*), III (*cob*), IV (*cox1*), and V (*atp6-1*) of the electron transport chain, genes for cytochrome C biogenesis (*ccmC* and *ccmFc*), ribosomal and transport proteins (*rps4* and *mttB*), rRNA (*rrn26*), and the maturase gene (*matR*).

Some of the selected genes (*nad6*, *cox1*, *ccmC*, *mttB*, *rps4*, and *matR*) were preferentially transcribed by RPOTmp, and their transcription levels were reduced in *rpotmp-2* (Figure 1, Appendix A), which is consistent with the data obtained by Kuhn et al. [5]. The steady-state levels of some RPOTmp-independent transcripts (*rnn26*, *ccMFc*, and *nad3*) were even enhanced in the absence of RPOTmp. These alterations are thought to be associated with elevated levels of cellular mtDNA caused by general energy constraints in the mutant [5].

According to the results obtained, all selected mitochondria-encoded genes were strongly repressed by MJ and induced by CK (except for *rrn26*) in the wild-type (WT) seedlings (Figure 1, Appendix A). The response to other hormones was gene-specific, with certain mt genes exhibiting expression shifts, but others remaining unaltered. Thus, *cox1* and *nad6* displayed a 2-fold increase in transcript abundance following the BL, IAA, ACC, GA_3_, and SA treatments, while none of these hormones affected *atp6* expression. The *rrn4* gene was upregulated by BL and IAA, but induction was not observed in response to ACC, GA_3_, or SA. In this regard, it should be noted that some hormone responses may be near saturation due to optimal endogenous concentrations.

Another potential complication in assessing the sensitivity of mt-encoded genes to hormones is associated with the phases of ontogenesis. Depending on the age of a plant, gene expression changes resulting in either an increase or decrease in the transcript levels can be triggered by the same treatment. Thus, the expression of mt-encoded genes sharply decreased when the wild-type seedlings were grown in the dark for 4 days on a medium containing *trans*-zeatin (CK, 1 µM) compared with the seedlings that were cultivated on a medium without a hormone (Appendix A). Hence, the changes in mt gene expression in response to hormonal treatment may represent the outcome of completely different effects, reflecting opposite pathways of their regulation.

### 2.2. Disruption of RPOTmp Alters Sensitivity to Hormone Treatment

Changes in the expression of nuclear-encoded RNA polymerases (NEP) through direct promoter-binding interactions with hormone-dependent transcription factors may affect the transcription of mitochondrial genes. To test the putative role of RPOTmp in hormone-dependent mitochondrial transcription, we analyzed the accumulation of the mitochondrial transcripts in Columbia 0 and RPOTmp-deficient mutant *rpotmp-2*.

Shifts in the transcript abundance of mt-encoded genes were abolished in the *rpotmp-2* background following the hormone treatment, with the exception of the BL-induced accumulation of *mttB* matrices (Figure 1). Moreover, several genes that were upregulated in the WT were even repressed in *rpotmp-2.* It thus appears that the loss of RPOTmp blocks or attenuates the hormone-dependent responses of mt-encoded genes. Strikingly, similar responses were observed for both RPOTmp-dependent and RPOTmp-independent genes when the magnitudes of their fluctuations in the mutant were assessed to be relative to the basal values. Moreover, such a response was also found for the hormones, the corresponding regulatory elements of which were absent in the promoters of the mt polymerases. These results suggest that the modulation in mt gene regulation may be the result of an altered hormonal status of *rpotmp-2*. The changes may also reflect a modified expression in the mutant of the second mitochondrial polymerase, RPOTm, which is the only active one in the *rpotmp-2* mitochondria, since hormone-related changes in the *RPOTm* transcript levels were mitigated in the *rpotmp* background (Figure 2).

To experimentally evaluate the sensitivity of *rpotmp-2* to hormone treatment, we first examined the hormone reporters. The mutant and WT had similar levels of the auxin marker *IAA19* and the BS marker *DWF4* in the 2-week-old seedlings. At the same time, the mutant exhibited elevated steady-state expression values of ethylene and the MJ marker *ERF1*. In parallel, the *rpotmp-2* mutant was less sensitive to ACC in root elongation and dark detached assays than WT (Figure 3). We further found that the mutant leaves were less sensitive to MJ in the dark detached leaf assay, which was consistent with a higher transcript abundance of the MJ marker gene *ERF1* (Figure 3). In contrast, the expression levels of the ABA reporter, *RD29*, were reduced in *rpotmp-2* and were two times more upregulated by the hormone than those in the wild type. The mutant was more sensitive to the ABA treatment in the root elongation assay, especially in the germination tests (Figure 3), and had expectedly reduced levels of the gibberellic acid synthesis gene, *GA3.*

In addition, we found that the mutant exhibited higher expression levels of the CK marker gene *ARR5* and the SA reporter *PR1*, suggesting a possibly elevated content of corresponding hormones. However, there were no significant differences in their responses to the CK and SA treatments in the physiological tests (Figure 3).

In summary, it can be concluded that the disruption of *RPOTmp* may alter the hormonal status of the mutant and its response to a treatment with exogenous hormones. Furthermore, the changes in the mt transcript levels in the mutant background may be the result of multifactor events, when impaired RPOTmp function and altered hormonal metabolism are superimposed on the modified RPOTm activity and, possibly, on the activity of additional transcription factors that bind directly to the promoter regions of mitochondrial genes.

### 2.3. The RPOTmp Promoter Has Potential Cis-Regulatory Elements That Respond to Phytohormones

An in silico analysis highlighted a number of consensus sequences in the 1.2 kb promoter region upstream of the RPOTmp coding sequence recognized by potential *cis*-regulatory elements (CREs) that may be involved in the response to phytohormones. The most significant differences are listed in Table 1. The putative motifs at positions −297/−287 bp and −1081/−1073 bp (AGATCCTC) and −966/−958 bp (AAAGATTCGA) relative to ATG (Appendix A) are well aligned with the consensus sequence 5′-(AGATHY, H(a/t/c), Y(t/c))-3′ [14] for cytokinin-sensitive type B response regulators (ARR-B) in a direct strand. Reverse complement sequences of ARR-B (TCGAATCTTT and GAGGATCTTA) were also found.

Two to four putative auxin-responsive elements (AuxRE) were predicted within the *RpoTmp* promoter. However, only two of them, GGGTCGGGTA for ARF3 at position −305/−315 bp in the direct strand and TCAGACAAAA for ARF5 at −799/−808 bp in the complementary strand, contained the canonical motif AuxREs 5′-(TGTCNC, N(a/t/c/g))-3′ for the auxin-responsive factor (ARF) [15].

We did not detect the classical G-box with ABA-responsive elements: ABRE 5′-((c/g/t)ACGTG(g/t)(a/c))-3′ and coupling element 3 (CE3; ACGCGTGTC), characteristic of ABA-regulated genes. At the same time, a 1.2 kb region of *RPOTmp pro* is abundant with *cis*-regulatory elements for ABA-regulated genes, including the MYB (5′-c/tAACNA/G-3′), MYC (5′-CANNTG-3′), WRKY (5′-(T)(T)TGAC(C/T)-3′), and RAV (5′-CAACA-3′; 5′-CACCTG-3′) family transcription factors (Table 1) [16]. In parallel, the DPBF1&2 binding site motif at −923/−927 bp (ACACCTG) in a complementary strand could be indirectly responsible for the reactions to the ABA treatment [17].

The 5′UTR of *RPOTmp pro* also contains various CREs (Appendix A), including two sequences specific for REF6 (RELATIVE OF ELF6) with the consensus motif 5′-CTCTGYTY-3′, which may play a role in ethylene and brassinosteroid signaling, and in ethylene response elements (EREs) or GCC boxes with the 5′-GCCGCCGCC-3′ core sequence [18].

In addition, the cross-regulation of the *RPOTmp* gene expression by different phytohormones can be achieved through numerous MYB or MYB-related factors even in the absence of characteristic CREs (Appendix A).

### 2.4. Hormone Treatments Induce Changes in GUS Expression in the RPOmp::GUS Reporter Strain

To assess the functional properties of the identified motifs, a genetic construct based on the pCAMBIA1381z vector containing a 5′-deletion fragment (−958 bp) upstream of the start codon of *RPOTmp* was fused to the open reading frame of the β-glucuronidase reporter gene and used to generate the *RPOTmp pro::GUS* line (Figure 4A). The β-glucuronidase staining of 10-day-old seedlings of transgenic P958 (T2 and T3 generations) grown on half MS showed stable blue coloration in the vascular tissues of the roots, cotyledons, and primary leaves as a result of *RPOTmp pro::GUS* expression (Figure 4B), while plants expressing the reporter *GUS* gene without a promoter remained virtually unchanged.

To examine whether the expression patterns of the construct change in a hormone-dependent manner, 10-day-old plants were treated with solutions of hormones for 24 h at 22 °C under a 16 h light regime, and GUS staining was performed. Plants expressing GUS activity under the 35S CAMV promoter were used as positive controls, whereas plants expressing the reporter *GUS* gene without a promoter were used as negative controls (Figure 5).

We found that GUS staining was obviously darker after the ACC and IAA treatments in plants containing the −958 bp fragment than in the control samples. There were also no pronounced differences from the control variants when the reporter strain was treated with CK, GA, BL, or SA. In parallel, the signal decreased when the seedlings containing the 958 bp fragment were exposed to ABA. From these observations, we conclude that the −958 bp promoter fragment of the *RPOTmp* gene is functionally involved in responses to IAA, ACC, and ABA.

The GUS activity in the *RPOTmp::GUS* reporter strain only partially corresponded to the *RPOTmp* transcript abundance under the hormone treatment of the WT plants. According to our qRT–PCR tests, *RPOTmp*, as well as *RPOTm*, were reproducibly induced by CK, IAA, ACC, and BL and downregulated by ABA in the WT plants (Figure 2). They were also downregulated by SA, although no reliable changes in the GUS activity were observed when the *RPOmp::GUS* reporter strains were treated with these hormones.

We therefore conclude that changes in the expression of *RPOTmp* may affect the transcription of mitochondrial genes both directly, through promoter-binding interactions with hormone-dependent transcription factors, and indirectly.

### 2.5. Exogenous Hormone Treatment Modulated the MTERF and SWIB Family Genes

Hormone-related shifts in the expression of organellar genes can also be induced by organellar-specific transcription factors via direct binding to the transcription initiation sites of mt genes. Proteomic studies revealed the presence of a large number of proteins containing DNA-binding motifs in plant mitochondria, which are expected to play key roles in mtDNA expression [19]. Some of these factors may represent facilities for the hormonal regulation of mt gene expression. Among them is a diversified mTERF family that includes 35 members targeted to chloroplasts and/or mitochondria, where they have been shown to function in OGE at the transcriptional or posttranscriptional level [20]. Although the members of the “mitochondrial” and “mitochondrion-associated” clusters respond weakly to physiological perturbations [10], at least some of them were up- or downregulated by more than 2-fold in response to exogeneous hormones according to the microarray data provided on the resource server (http://bar.utoronto.ca/efp/cgi-bin/efpWeb.cgi (accessed on 2 October 2023)).

To test whether they could represent instruments for the transduction of hormonal signals to mitochondria, we focused on mTERF15, 17, and 18 since their promoter regions are predicted to possess binding site motifs for hormone-regulated transcription factors (https://agris-knowledgebase.org (accessed on 2 October 2023)). mTERF15, with an experimentally confirmed function, is naturally induced during germination [10] and is needed for *nad2* intron 3 splicing [21]. Its dysfunction disrupts formation and decreases its activity of complex I. The MTERF18 transcript levels were shown to fall upon the exposure to heat and during germination [10].

In 10-day-old light-grown seedlings, the transcripts of all three genes of interest increased in abundance by by 2 to 4-fold in response to IAA or CK and decreased by approximately 2–5 times after treatments with ABA, SA, and MJ. This finding correlates well with the data obtained for *RPOTm* and *RPOTmp* and with the hormone-mediated expression of mt-encoded genes. It is worth noting that while *mTERF15* and *18* were upregulated by ACC, *mTERF17* was even slightly repressed. A unique response of *mTERF17* was also observed in the etiolated seedlings grown on a medium containing CK. While its expression was upregulated, the expressions of *mTERF15* and *18* were suppressed in accordance with the repressive regulation of mt-encoded genes by cytokinin in this experimental setup. These results suggest that members of the mTERF family may have overlapping but also specific functions in the hormone-mediated regulation of mt-encoded genes depending on the development status and/or environmental conditions.

The effects of hormonal application could also be regulated through tetra- and pentatricopeptide repeat proteins (TPRs and PPRs) targeted to mitochondria and coexpressed with mTERFs of the mitochondrial cluster. We have shown that two such genes, *At1g09190* and *At2g37320*, encoding TPR-like superfamily proteins, followed *mTERF15* and *mTERF18* in their hormone-mediated expression patterns (Figure 6, Appendix A) and corresponded to the expression patterns of some mt-encoded genes.

In addition, modulations in the expression of mt-encoded genes could be attributed to the activity of the mitochondrial nucleoid-associated protein, SWIB5, a member of the SWIB (ATP-dependent multisubunit switch/sucrose nonfermentable multiprotein complex B) family, which is implicated in DNA binding and remodeling. SWIB5 associates with mtDNA and participates in the regulation of mitochondrial gene expression [11]. According to our tests, the relative expression values of *SWIB5* were increased by 3–10-fold after CK, ACC, and IAA applications and increased by nearly 6-fold following the BL and GA_3_ treatments (Figure 6). Interestingly, the responses of the mTERF genes and the genes for the mt RNA polymerases to the last two hormones were considerably weaker and barely exceeded 1.5 times despite the significant induction of some mt-encoded genes.

As expected, stress-related hormones (ABA, MJ, and SA) downregulated all aforementioned nuclear genes. However, it is worth noting that several mt-encoded genes (*nad6*, *mttB*, *matR*, *cob*, and *ccMc*) were significantly activated by the ABA treatment, suggesting the involvement of alternative regulatory pathways.

In summary, these data indicate a function for MTERF and TPR proteins coexpressed with MTERF in the hormone-regulated expression of mt-encoded genes in addition to organelle RNA polymerases and mitochondrial nucleoid-associated proteins.

## 3. Discussion

In general, the obtained results indicate that the hormone-mediated responses of mt genes in Arabidopsis seedlings can be attributed to the activities of mitochondrial RNA polymerases that are able to bind hormone-dependent transcription factors and, possibly, to supplementary transcription factors that can bind directly to the promoter regions of mitochondrial genes. This assumption is at least partially supported by hormone-induced changes in GUS expression in the *RPOmp::GUS* reporter strain and the coordinated transcription responses of *RPOTmp* and/or *RPOTm* with some mitochondrial-encoded genes following hormone treatment. The most regular fluctuations occurred upon the treatments with CK and MJ, which promoted the up- or downregulation of all tested mt genes in a context-specific manner. The responses of the selected mt genes to other hormones were gene-specific and did not always follow the expression patterns of the RNA polymerase genes, which is consistent with the idea that nuclear and mitochondrial transcriptions may be independently regulated [22].

In particular, a side-by-side comparison showed that the downregulation of *RPOTmp* by ABA corresponded to the upregulation of a number of mt genes, including *nad6*, *cob*, *ccMc*, *ccMfc*, and *mttB* (Figure 1 and Figure 2). The enhanced mitochondrial activity induced by the ABA treatment may be a consequence of increased energy consumption in response to stressful situations, which are usually accompanied by an increase in ABA levels. The growth effects documented for the *RPOTmp* transcripts following the IAA treatment were not observed for *atp6* or *rnn 26*, although the expressions of other selected mt genes were significantly increased. This specificity could provide a means to fine-tune the activities of certain genes in response to multiple challenges.

While our tests clearly linked the CK-induced changes in *RPOTmp* and *RPOTm* transcript abundance to the modulations of mt-encoded gene expression, there was one unexpected finding. The GUS activity in the *RPOmp::GUS* reporter strains did not show changes following the CK treatment, contrary to the upregulation of *RPOTmp* transcript abundance. Such results clearly contradicted the presence of putative CK-regulated *cis*-elements in the *RPOTmp* promoter, as predicted via bioinformatic resources. It is noteworthy, however, that the accessibility of such binding sites is doubted by the ChiP studies of Zubo et al. [23] and Xie et al. [14], who did not find binding locations for type B ARRs in the *RPOTmp* promoter. Overall, these data indicate that the CK-dependent regulation of *RPOTmp* may be indirect.

It should be noted as well that the lack of visible changes in the GUS expression patterns may also be caused by a low promoter usage and does not necessarily correspond to the level of transcript accumulation. As stated by Lieber et al. [24], “These two technologies focus on very different steps in gene expression (i.e., promoter usage versus transcript accumulation) and differences between them can be explained by, selective transcript stabilization or reduced transcript degradation”.

In addition to mt RNA polymerases, transcription factors directly binding to the promoter regions of mitochondrial genes could be implicated in mediating phytohormone signals to mt genes. Among them, the genes of the mTERF family and the genes for the tetra- and pentatricopeptide repeat proteins (TPRs and PPRs) coexpressed with the *mTERF* genes are of particular interest since they are regulated by hormone treatment and are capable of binding nucleic acids [20]. In our tests, the MTERF and TPR protein genes specifically responded to the hormone treatment, displaying both negative and positive regulations in a context-dependent manner. According to bioinformatic resources, the promoter regions of these genes possess putative *cis*-acting elements involved in the responses to a number of phytohormones, including ABF, ABRE, GATA, W-box, etc. (Agris). However, additional experiments are needed to confirm the physical interactions between MTERFs and hormone-induced TFs. Of note, mTERF17 and 18 have been shown to bind type B response regulators ARR12 and ARR1,10, and 12, respectively [14], thus presenting direct targets for cytokinin signaling.

Binding motifs for ARR1 and 10 were also revealed in the promoter region of *SWIB5* [14,23]. In our experiments, this gene was strongly upregulated by CK in light-grown seedlings and downregulated in etiolated seedlings grown on a CK-containing medium in the dark, which was consistent with the changes in the transcript abundances of the selected mt-encoded genes. The protein encoded by *SWIB5* belongs to the nucleosome remodeling complex of mitochondria, similar to the bacterial nucleoid. It is essential for cell proliferation and stress responses and is believed to adjust the accessibility of mtDNA for RNA polymerases, linking hormone responses with chromatin remodeling [25]. SWIB5OE displayed a significant downregulation of *CRF6* (CYTOKININ RESPONSE FACTOR6) [11], encoding a cytokinin-responsive AP2/ERF transcription factor that plays a key role in the inhibition of dark-induced senescence and oxidative stress as a negative regulator of the CK-associated module. [26]. Furthermore, CRF6 refers to a stimulator of mitochondrial dysfunction (MDS) induced by mitochondrial perturbation. Since both cellular proliferation and stress responses are associated with CK-mediated modulations, we suggest that the CK-dependent regulation of SWIB5 may be one of the mechanisms underlying the expression of mitochondrial genes. Additionally, the involvement of SWIB5 in responses to several other plant hormones suggests pleiotropic functions in the regulation of the mt genome. However, this suggestion must be further addressed in future experiments.

Since any biological function is usually implemented by several independent mechanisms, both organelle RNA polymerases and mitochondrial transcription factors, as well as mitochondrial nucleoid-associated proteins acting redundantly, can be direct targets for hormone-regulated transcription factors. They can form a core regulatory module that acts in the direct transduction of hormonal signals to mitochondria. Alternatively, the hormone-related transcriptional activity of these genes may be modulated indirectly, suggesting that additional factors are needed for their regulation. This is especially relevant for brassinosteroids, since the promoters of *RPOTmp* (and of selected genes for mTERF and PPR proteins) do not contain consensus binding sites for the BS-induced transcription factors BZR and BES.

Strikingly, the upregulation of mt genes by BS (except for *mttB*) was dampened in the *rpotmp-2* background in the same way as for the hormones whose transcription factors can directly interact with the *RPOTmp* promoter. Therefore, the loss of RPOTmp has far-reaching implications for the activity of hormone-related genes. According to the data obtained by Meredino et al. [1], the impaired function of RPOTmp caused changes reminiscent of the triple response of seedlings exposed to ethylene and could therefore contain increased levels of ethylene. In accordance with extensive crosstalk and signal integration among growth-regulating hormones, plants with a reduced or increased content of one hormone can show altered responses to another [27]. Thus, *rpotmp* exhibited increased steady-state levels of transcripts for the CK marker gene, ARR5, as well as an elevated transcript accumulation of the CK synthesis genes *IPT3* and *IPT5* and a reduced level of the hormone catabolism gene *CKX3* expression, which implies a possible increase in the content of endogenous cytokinins [28]. In parallel, the expressions of reporter genes for GA (*GA3*) and SA (*PR1*) were changed in the *rpotmp* compared to the WT (Figure 3). These results suggest that the disruption of RPOTmp may induce a hormonal imbalance in concerted hormonal synthesis and signaling and, as a result, a differential mitochondrial response to hormonal treatment. It thus appears that the knockout or overexpression of genes regulating organellar proteins can provoke indirect effects that cast doubt on the validity of corresponding mutants in the elucidation of naturally occurring hormone-dependent responses.

On the other hand, a change in the hormonal status of the *rpotmp* mutant may be a consequence of retrograde signaling from dysfunctional mitochondria. It has been suggested that ethylene boosts mitochondrial respiration and restores mitochondrial function upon mitochondrial proteotoxic stress (mitochondrial unfolded protein response) as the anterograde arm of a feedback loop [4]. This was accompanied by MAPK6 activation and an increase in the transcription of the ethylene synthesis gene *ACS6*. Similarly, altered ethylene levels in the *rpotmp* mutant may be a means to recover mitochondrial functionality under reduced levels of respiratory complexes I and IV.

Other hormone responses to mitochondrial proteotoxic stress included the induction of auxin, ABA, and jasmonate signaling; a decrease in cytokinin signaling; and unchanged salicylic acid signaling [4]. Notably, in line with these results, detached *rpotmp* leaves were more resistant to the MJ treatment (Figure 3C), suggesting increased steady-state levels of this hormone in the case of mitochondrial dysfunction.

The role of auxin is especially significant. Two independent works revealed that mitochondrial stress stimuli caused a suppression of auxin signaling, and conversely, auxin treatment repressed mitochondrial stress [29,30]. According to the results of our analyses, ethylene and IAA reproducibly induced the transcript accumulation of *RPOTmp* and RPOTmp::GUS fusion activity, which correlated with the enhanced levels of mitochondrial RNAs. This implies a direct signaling interaction between these two hormones and RNA polymerase in the transduction of hormone operational signals from the nuclei to mitochondria.

In general, the results of this study indicate that hormones are essential mediators that regulate mitochondrial gene expression in a context-dependent manner. Inputs from multiple hormones can cause/induce alterations in the transcript accumulation of mt-related nuclear genes, which, in turn, trigger the expression of mt genes.

## 4. Materials and Methods

### 4.1. Plant Material, Growth Conditions, and Hormone Treatment

Arabidopsis (*Arabidopsis thaliana*) ecotype Columbia 0 and *rpotmp-2* (NASC N6328420; Salk 132842) were used in our experiments. *RPOTmp::GUS* and *35S::GUS* were generated in this work. The seeds were soaked in a 30% bleach solution for 10 min, rinsed 3 times with sterilized water, and stratified for 2 days in darkness. Seedlings were grown in half-strength Murashige and Skoog (MS) medium (pH 5.7) containing 1% sucrose and 0.5% phyto agar (Duchefa Biochemie, Haarlem, The Netherlands) in a growth chamber at 22 °C with a 16 h light/8 h dark cycle. Ten-day-old seedlings were treated with the hormone solutions and collected after 3 h of exposure, unless otherwise stated. The compounds assayed included abscisic acid (ABA, 5 × 10^−5^ M), gibberellic acid (GA_3_, 10^−6^ M), indole-3-acetic acid (IAA, auxin, 10^−6^ M), 1-aminocyclopropane-1-carboxylic acid (ACC; ethylene precursor 10^−5^ M), *trans*-zeatin (CK; 5 × 10^−6^ M), brassinolide (BL, 10^−7^ M), salicylic acid (SA, 10^−5^ M), and methyl jasmonate (MJ, 5 × 10^−5^ M), in addition to a mock treatment. The concentrations of active reagents and treatment time were selected in preliminary experiments. The effectiveness of hormonal treatment was confirmed via an expression analysis of marker genes specific for each hormone.

For tests with mature plants, the seedlings were transferred into the soil and grown until the age of 5 weeks. CK-dependent effects were also studied in a model system designed by Cortleven et al. [31]. Seeds were germinated in the dark for 4 days on full MS medium with or without cytokinin (1 µM *trans*-zeatin).

### 4.2. Hormone Sensitivity Assays

For the seed germination assay, seeds were grown on MS medium supplemented with different concentrations (0, 1, 2, 3, or 6 µM) of ABA at 22 °C with a 16 h light/8 h dark photoperiod. The number of fully expanded cotyledons was estimated after 6 days in triplicate with 50 seeds for each experiment.

For the root elongation assay, 10-day-old seedlings were transferred to ½ MS plates with different concentrations of phytohormones and grown vertically for another 4 days under 16 h light/8 h dark conditions. Measurements were performed in triplicate with 20 seedlings for each experiment.

A chlorophyll retention assay was performed with the 5th and 6th rosette leaves excised from the soil-grown plants. The leaves were placed on filters moistened with hormone solutions or water with a solvent and kept for 3 days in darkness. The total chlorophyll content (chlorophyll a and chlorophyll b) was measured as described by Lichtenthaler [32] and was related to the leaf area (mg/cm^2^). Three samples each containing 3 leaf discs were measured for each test. The chlorophyll content at the start of the experiment was taken as a reference and set at 100%.

### 4.3. RNA Extraction and Quantitative RT–PCR (qRT–PCR)

Total RNA from hormone-treated and control 10-d-old seedlings was isolated via extraction using TRIzol reagent (Thermo Fisher Scientific, Waltham, MA, USA). RNA content was determined using a spectrophotometer Nanodrop ND-1000 (NanoDrop Technologies, Wilmington, DE, USA). DNA contamination was removed with DNAse I (Thermo Fisher Scientific) treatment. cDNA synthesis was performed using RevertAid Reverse Transcriptase (Thermo Fisher Scientific) and a mixture ofoligo-dT_12-18_ and random primers (DNA synthesis, Moscow, Russia). RT–PCR was performed using a LightCycler 96 (Roche, Basel, Switzerland) with the qPCRmix-HS SYBR + LowROX system (Eurogen, Moscow, Russia). The qPCR program consisted of 1 cycle at 95 °C for 5 min to activate the polymerase, which was followed by 40 cycles of 15 s at 95 °C, 15 s at 58 °C, 25 s at 72 °C, and melting curve analysis [33]. Each reaction was performed in triplicate. To verify the specificity of the DNA amplification products, a dissociation curve was analyzed for each sample. The nucleotide sequences of primers for qRT–PCR analysis are given in Appendix A. *UBQ10* (*At5g53300*) and protein phosphatase 2A (PP2A) regulatory subunit A2 (*At3g25800*) genes were used as internal controls for qRT–PCR data normalization.

### 4.4. Prediction of Cis-Acting Elements in the RpoTmp Promoter

Using the bioinformatics resources of the Arabidopsis Information Resource (TAIR; https://www.arabidopsis.org/ (accessed on 12 September 2023)), the promoter sequence of *RPOTmp* (1200 bp upstream of the translation initiation site ATG) was obtained. The *cis*-acting elements were predicted using plant *cis*-acting regulatory DNA elements (PLACE, http://www.dna.affrc.go.jp/htdocs/PLACE/ (accessed on 1 October 2023)) [34]; Plant CARE (http://bioinformatics.psb.ugent.be/webtools/plantcare/html/ (accessed on 10 October 2023)) [35]; PlantRegMap/PlantTFDB v5.0 (http://plantregmap.gao-lab.org/binding_site_prediction.php (accessed on 10 October 2023)) [36]; and the Arabidopsis Gene Regulatory Information Server (AGRIS) (https://agris-knowledgebase.org (accessed on 20 August 2023)) [37] databases. The gene transcription start site (TSS) and promoter region were indicated using the TSSPlant and TSS tools PlantProm DB database [38] (accessed on 10 October 2023). Nucleotide sequence analysis, primer design, and localization of regulatory elements were performed with Vector NTI Advance 9.0 (Invitrogen, Waltham, MA, USA) and visualized (Appendix A). The results of the search obtained with PlantRegMap are summarized in Appendix A**.**

### 4.5. Construction of the RPOTmp Promoter::GUS Reporter Gene Fusion and Agrobacterium Transformation

Arabidopsis genomic DNA was extracted using the CTAB method. A 958 bp (−958/−1) upstream of the translation start codon of *RPOTmp* was amplified from the genomic DNA via polymerase chain reaction (PCR) using forward and reverse primers carrying BamHI and PstI restriction sites, respectively. The primers used for plasmid construction and sequencing are listed in Appendix A. The resulting BamHI/PstI fragment was cloned upstream of the promoterless *uidA* reporter gene (*GUS*) of the binary vector pCambia-1381Z (Clontech, Mountain View, CA, USA), producing pCambia *RPOTmp pro*::*GUS* construct, namely, P958. After sequencing to exclude possible point mutations, the resulting construct was introduced into *Agrobacterium tumefaciens* strain GV3101 (C58) (GoldBio, St Louis, MO, USA) using the freeze–thaw method [39]. Positively transformed *A. tumefaciens* cells resistant to kanamycin (50 µg/µL) and rifampicin (100 µg/µL) were selected on LB plates at 28 °C for two days.

### 4.6. Generation of RPOTmp pro::GUS Arabidopsis Transgenic Plants

*A. tumefaciens* cells harboring P958 plasmids were used to transfect 4- to 5-week-old wild-type Arabidopsis plants in the Col-0 background using the floral dip method in the presence of 5% sucrose and 0.01% (*v/v*) Silwet L-77, as previously described by Clough and Bent [40]. The original genetic constructs of pCambia1381Z and pCambia1301-35S (Clontech, Mountain View, CA, USA) were used to generate negative and positive control plants, respectively, based on GUS activity. T1 transgenic seeds from each transformant plant were tested for germination on half MS medium containing 0.8% (*w/v*) phyto agar (Duchefa Biochemie) and 30 mg/L hygromycin (Hyg). The Hyg-resistant seedlings were then grown in soil or on ½ MS medium at 22 °C under long-day conditions for further analysis (Appendix A). To improve the integration of the P958 construct into Col-0 plants, genomic DNA was extracted from rosette leaves of transgenic T1 plants and used for PCR analysis with *GUS*-specific primers (Appendix A). The T2 and T3 generations were used for the subsequent analysis of GUS activity.

### 4.7. GUS Staining

GUS activity was assayed in *Hyg*-resistant P958 seedlings after 24 h of incubation in half MS supplemented with appropriate phytohormone or without it (control) at 22 °C with 16 h daylight. All treatments were performed in similar time and daylight periods. Seedlings of the pCambia1381Z and pCambia1301-35 transgenic lines were used as control plants. GUS staining was performed according to Gallagher [41]. Samples were collected and immediately fixed in ice-cold acetone (90%) for 30 min at 4 °C in the dark, rinsed 2 times for 15 min with 50 mM phosphate buffer (pH 7.0), and immersed in a GUS assay solution containing 0.5 mM X-Gluc (5-bromo-4-chloro-3-indoly-β-D-GlcA) in 50 mM phosphate buffer, 0.1% (*v/v*) Triton X-100, 1 mM EDTA, 0.5 mM potassium ferrocyanide, and 0.5 mM potassium ferricyanide. The staining was carried out between 4 and 16 h at 37 °C. GUS-colored tissues were washed gradually with 25 to 75% (*v/v*) ethanol to remove chlorophyll. The results were recorded photographically using a stereomicroscope MSP-2 (LOMO, St. Petersburg, Russia).

### 4.8. Statistical Data Processing

All experiments were performed in three biological replicates. The data were evaluated via one-way analysis of variance (ANOVA) followed by Tukey’s method using an online calculator (astatsa.com/OneWay_Anova_with_Tukey HSD/ (accessed on 10 October 2023)). All data are presented as the means ± their standard errors (SE).

## 5. Conclusions

Dynamic communication between the nucleus and mitochondria requires the acute coordination of various aspects of their activities in which phytohormones play vital roles. This process is predominantly regulated by the nuclear genome, as over the course of symbiotic evolution, the majority of mitochondrial genes migrate into the nuclear genome, leaving a set of essential genes that encode the subunits of the respiratory chain, heme and cytochrome assembly, and mitochondrial ribosomes. Here, we have shown that the hormone-related expression of the mitochondrial genome is at least in part regulated via the genes encoding nuclear-encoded RNA polymerases, RPOTmp and RPOTm, as well as MTERF and SWIB family members. According to our RT–PCR tests, these genes are reproducibly induced by CK, IAA, ACC, and BL and downregulated by ABA in WT plants, modulating the transcript accumulation of mt genes in a context-dependent manner. The disruption of *RPOTmp* blocks or attenuates the hormone-dependent responses of mt-encoded genes and alters the sensitivity of the *rpotmp-2* mutant to hormone treatment. It should be noted, however, that the mutation may induce a hormonal imbalance in concerted hormonal synthesis and signaling and, as a result, a differential mitochondrial response to hormonal treatment. An in silico analysis highlighted a number of consensus sequences in the regions of the selected nuclear-encoded mt genes recognized by *cis*-regulatory elements that may be involved in the response to phytohormones. Moreover, the functional properties of the potential motifs were verified with a construct possessing the *GUS* gene fused to the −958 bp promoter fragment of the *RPOTmp* gene. The GUS activity in the *RPOTmp::GUS* reporter strain was upregulated in response to IAA and ACC and downregulated by ABA. In parallel, hormone-related changes in the transcriptional activity of mt genes may be modulated indirectly, suggesting that additional factors are needed for their regulation.

## Figures and Tables

**Figure 1 ijms-24-16924-f001:**
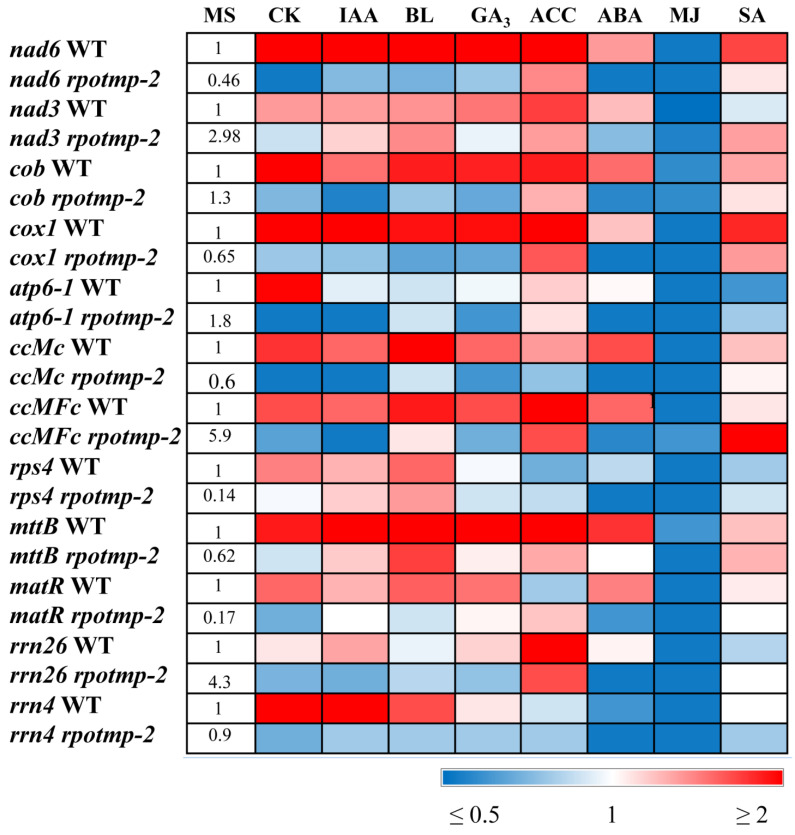
Effect of phytohormone treatment on relative gene expression values of mitochondrial genes. *A. thaliana* wild type and *rpotmp* mutant plants were grown on MS medium in Petri dishes at an illumination of 50 μmol·m^−2^·s^−1^ and a temperature of 23 °C with a 16 h photoperiod. Ten-day-old seedlings were treated with hormone solutions and collected after 3 h of exposure. Total RNA was isolated from seedlings and analyzed via relative quantitative RT–PCR using *UBQ10* and *PP2A* as internal standards. The numbers in the “MS” column indicate the baseline ratio of expression of each gene in the wild type and mutant without treatments. Full numerical data are presented in Appendix A.

**Figure 2 ijms-24-16924-f002:**
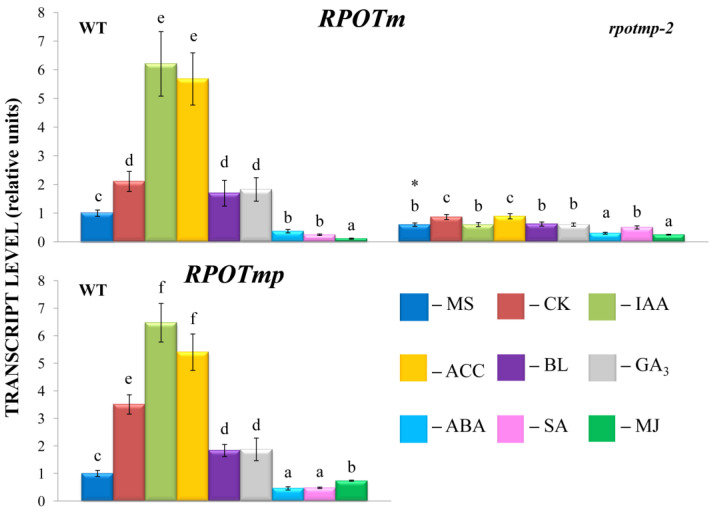
Effect of phytohormone treatment on relative gene expression values of nuclear-encoded page-type RNA polymerases RPOTmp and RPOTm. *A. thaliana* wild type and *rpotmp* mutant plants were grown on MS medium in Petri dishes at an illumination of 50 μmol·m^−2^·s^−1^ and a temperature of 23 °C with a 16 h photoperiod. Ten-day-old seedlings were treated with hormone solutions and collected after 3 h of exposure. Total RNA was isolated from seedlings and analyzed via relative quantitative RT–PCR using *UBQ10* and *PP2A* as internal standards. Different letters denote statistically significant differences among variants within the same genotype at *p* < 0.05 (ANOVA with Tukey’s post hoc multiple comparison test). Asterisk indicates statistically significant differences between the control variants of mutants and the wild type at *p* < 0.05 (*t*-test).

**Figure 3 ijms-24-16924-f003:**
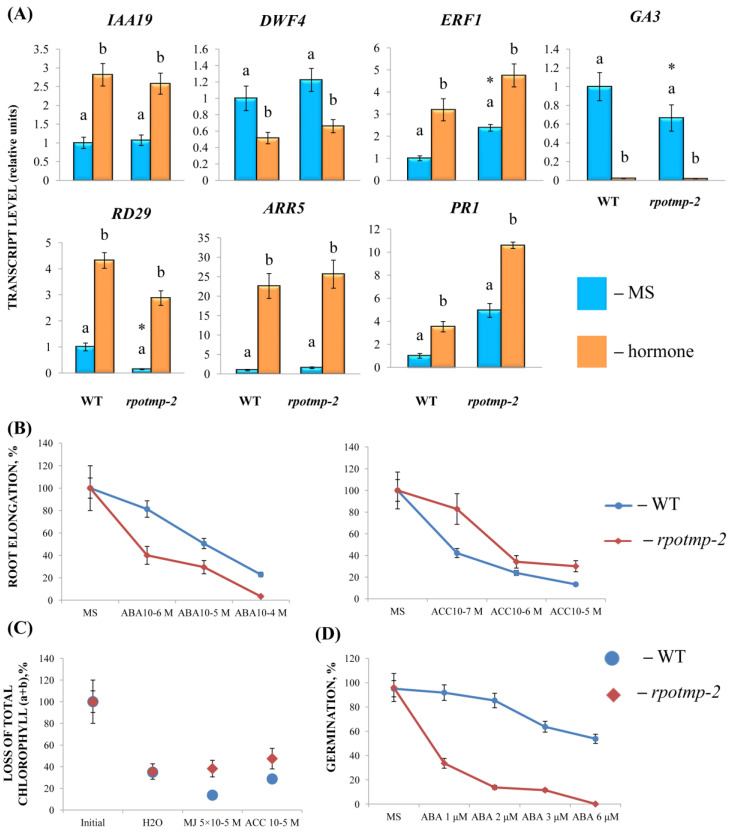
Sensitivity to hormone treatment of *A. thaliana* wild type and *rpotmp* mutant. (**A**) Relative expression values of phytohormone reporter genes. (**B**) Inhibition of root elongation via ABA and ACC treatments. Ten-day-old seedlings were transferred to 1/2 MS plates with a range of hormone concentrations and grown vertically for 4 days under 16 h light/8 h dark conditions. (**C**) Sensitivity to hormone treatment during dark-induced leaf senescence. Total chlorophyll content (chlorophyll a + chlorophyll b) was measured in detached 5th and 6th rosette leaves excised from the soil-grown 5-week-old plants and incubated in the dark at 23 °C on water, MJ (5 × 10^−5^), or ACC (10^−5^ M) solutions for 3 days. The loss of chlorophyll is presented as a percentage of the initial value. (**D**) Inhibition of seed germination via ABA. The percentage of fully expanded cotyledons was estimated after 6 days of germination on MS media with a range of ABA concentrations. Asterisk indicates statistically significant differences between the control variants of mutants and the wild type at *p* < 0.05 (*t*-test).

**Figure 4 ijms-24-16924-f004:**
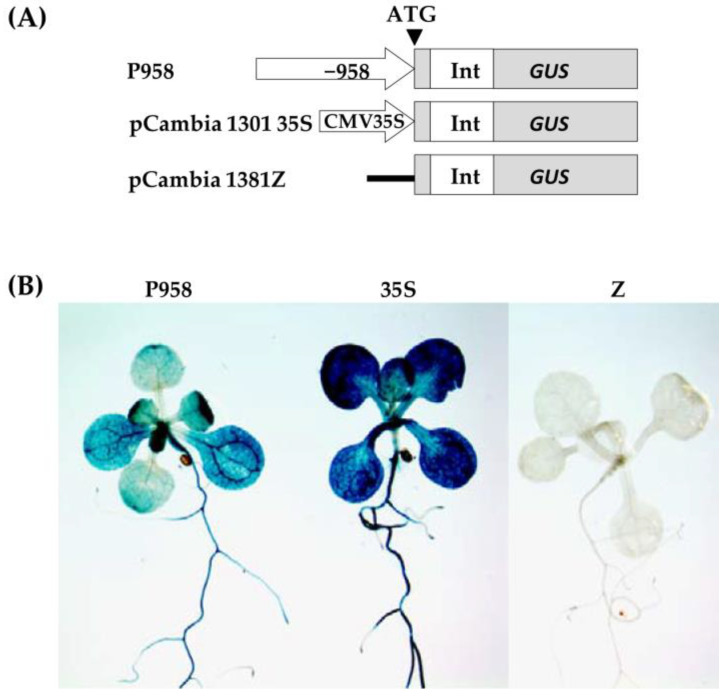
(**A**) Schematic of P958 genetic construct designed using the pCambia 1381Z plant expression vector for analyzing *GUS* reporter gene expression, where the translated region of the *GUS* gene is colored grey with start codon (ATG); Int-modified castor bean catalase intron within the translated region of the *GUS* gene; promoters are shown as arrows with corresponding signature (−958). pCambia1381Z and pCambia1301-35S are the original genetic constructs (Clontech, Mountain View, CA, USA) were used to generate control plants based on GUS activity. (**B**) GUS staining images of 2-week-old plants expressing GUS activity under the *RPOTmp* promoter containing a 5′-deletion fragment (−958 bp) upstream of the start codon (P958), under the 35S CAMV promoter (35S; positive control) and a negative control of the reporter *GUS* gene without a promoter (Z; negative control). Images were taken with a stereomicroscope, 0.6× magnification.

**Figure 5 ijms-24-16924-f005:**
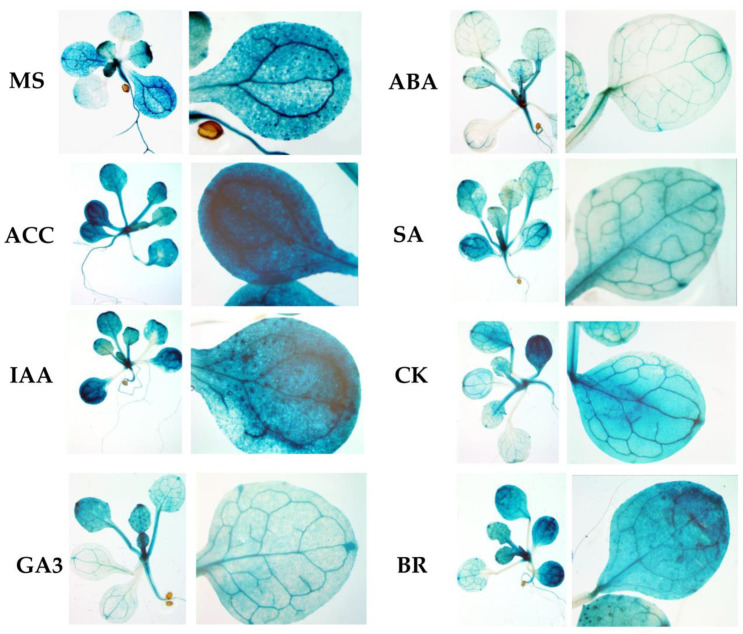
GUS staining images of 2-week-old plants expressing GUS activity under the *RPOTmp* promoter (P958) after 24 h of incubation in half MS supplemented with the appropriate phytohormone or without it (control) at 22 °C with a 16 h photoperiod. For each variant, images were taken at 10× (**left**) and 20× (**right**) magnification using a stereomicroscope.

**Figure 6 ijms-24-16924-f006:**
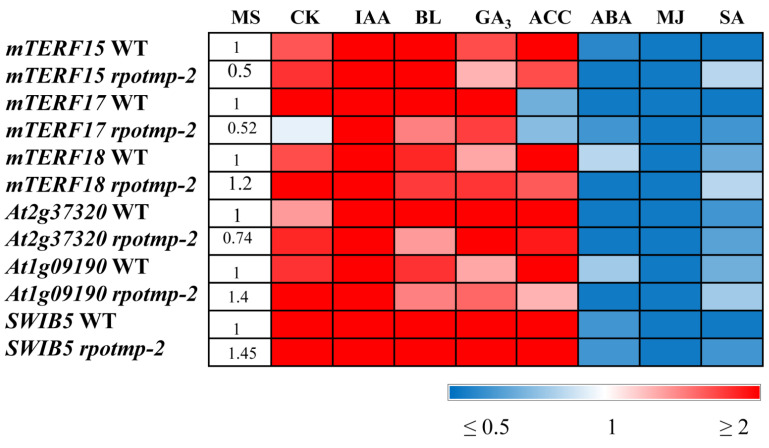
Effect of phytohormone treatment on relative gene expression values of nuclear-encoded genes. *A. thaliana* wild type and *rpotmp* mutant plants were grown on MS medium in Petri dishes at an illumination of 50 μmol·m^−2^·s^−1^ and a temperature of 23 °C with a 16 h photoperiod. Ten-day-old seedlings were treated with hormone solutions and collected after 3 h of exposure. Total RNA was isolated from seedlings and analyzed via relative quantitative RT–PCR using *UBQ10* and *PP2A* as internal standards. The numbers in the “MS” column indicate the baseline ratio of expression of each gene in the wild type and mutant without treatments. Full numerical data are presented in Appendix A.

**Table 1 ijms-24-16924-t001:** Putative *cis*-regulatory elements identified in the 1200 bp Arabidopsis *RPOTmp* promoter sequence relative to ATG in silico (based on the results of analysis using AGRIS, PlantRegMap/PlantTFDB v5.0, PLACE, and PlantCARE programs).

Putative *cis*-Regulatory Element	Consensus Motif	Sequence in *RPOTmp* (5′->3′)	Putative Transcriptional Factors Based on PlantRegMap/PlantTFDB v5.0
Auxin response factor Transcriptional factor ARF B3 family protein/ARF AUX/IAA-like protein	5′-(TGTCNC, N(a/t/c/g))-3′	TCAGACAAAA GGGTCGGGTA ATTGTTGACCAAAAAAATAAA	ARF5 (AT1G19850) ARF3 (AT2G33860) ARF16 (AT4G30080)
Cytokinin-responsive type B response regulators ARR-B	5′-(AGATHY, H(a/t/c), Y(t/c))-3′	AAAGATTCGA AGATCCTC GAGGATCTTA AGATTCGA	ARR11 (AT1G67710) ARR10 (AT4G31920) ARR2 (AT4G16110) ARR14 (AT2G01760)
ABA-responsive element AtMYC2 BS in RD22	5′-(CANNTG, N(a/t/c/g))-3′	CATGTG	
W-box promoter motif	5′-(T)(T)TGAC(C/T)-3′	ATTGTTGACCAAAAAA	WRKY2 (AT5G56270) WRKY20 (AT4G26640) WRKY23 (AT2G47260) WRKY38 (AT5G22570) WRKY60 (AT2G25000) WRKY62 (AT5G01900) WRKY63 (AT1G66600)
Related to ABI3 and VP1 (RAV1-A/RAV1-B) motif	5′-(CANNTG/A)-3′	CCAACGAAGATCACTCG	RAV1(AT1G13260)
DPBF1&2 binding site motif		ACACCTG	
Light-responsive elements			
SORLIP1AT	5′-(A/T)GATA(G/A)-3′	GCCAC	
EveningElement promoter motif	5′-C(A/C/G)ACA(N)_2–8_(C/A/T)ACCTG-3′	AAAATATCT	
GATA promoter motif	5′-AAAATATCT-3′	AGATAA TGATAG	
GCC-box	5′-GCCGCCGCC-3′	GGTTTAAGGCGGCTTCGTCGT	DREB2 (AT3G11020)

## Data Availability

Data is contained within the article and Appendix A.

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
