# Peer review of "Phytohormones as Regulators of Mitochondrial Gene Expression in Arabidopsis thaliana"

_ijms, 2023, doi:10.3390/ijms242316924_

Round 1

Reviewer 1 Report

Comments and Suggestions for Authors

Mitochondria and chloroplasts play a central role in the bioenergetics homeostasis of plants. Previous works indicate that the hormone-dependent expression of mitochondria (and chloroplasts) genes is determined by a nucleus-to-mitochondria anterograde signals. However, the molecular mechanism underpinning this process remains largely unknown, notably for mitochondria.

 The authors aims at understanding how changes in the hormone-induced expression of nuclear-encoded RNA polymerases (NEP) may alter the transcription of mitochondrial genes. 

It is shown that hormones induce the expression nuclear genes encoding RNA polymerases RPOTmp and RPOTm, which modulate the expression of the mitochondrial genes. In addition, it is shown that mitochondrial transcription factors (MTERF and SWIB family members) can also affect the mitochondrial gene transcription in response to hormones.

The manuscript is well written and the conclusion rests experimental data. The quality of the manuscript would have better if a subcellular localization of the RPOT transcription factors.                                                                                                                              

Comment.

Beginning of the Discussion.

‘In general, the results obtained indicate that the hormone-mediated responses of mt genes in Arabidopsis seedlings can be attributed to the activity of mitochondrial RNA polymerases able to bind hormone-dependent transcription factors and, possibly, to supplementary transcription factors that can bind directly to the promoter regions of mitochondrial genes. This assumption is at least partially supported by hormone-induced changes in GUS expression in the RPOmp::GUS reporter strain …’

The GUS results (Fig. 4-5) should be interpreted with caution. GUS experiments show the expression at a tissue level, the leaf. This does not proof that RPOTm (mp) are specifically target to mitochondria. Subcellular localization can be achieved using RPOT-GFP (and variants) fusion protein.

Author Response

We express our sincere gratitude to the reviewer who took the trouble to evaluate our manuscript. The comment to the made remark is given below

The GUS results (Fig. 4-5) should be interpreted with caution. GUS experiments show the expression at a tissue level, the leaf. This does not proof that RPOTm (mp) are specifically target to mitochondria. Subcellular localization can be achieved using RPOT-GFP (and variants) fusion protein.

Response

According to in vitro and in vivo import studies, a small family of three RPOT genes in Arabidopsis thaliana encodes a mitochondrial (RPOTm) and a plastidial (RPOTp) RNA polymerase and an enzyme imported into both mitochondria and plastids (RPOTmp) (Hedtke 2000, 2002). RpoTm has to be considered as the basic RNA polymerase in mitochondria of eudicots required for the transcription of most, if not all, mitochondrial genes (Kühn et al, 2009). In the RpoTmp mutants, decreased amounts of nad2, nad6 and cox1 transcripts and a lower abundance of the respiratory chain complexes I and IV were found suggesting a specific role of RpoTmp for the formation of these complexes. The statement that RPOTmp plays an important role in mitochondria, but not in chloroplasts, despite dual targeting, has been also proved by Tarasenko and colleges (2016). Its impact in chloroplasts is restricted to the recognition of the PC promoter in front of the 16S rRNA operon.

Reviewer 2 Report

Comments and Suggestions for Authors

Comments

  • Clarify the rationale behind the selection of specific hormones and their concentrations used in the experiments, ensuring they reflect physiologically relevant conditions in Arabidopsis.
  • Provide detailed methodological explanations, especially regarding the qRT‒PCR analysis and the criteria for classifying genes as "regulated" based on transcript change ratios.
  • Exercise caution in interpreting and generalizing results, particularly concerning the effect of hormone treatments on mitochondrial transcript abundance and the involvement of RPOTmp​.
  • Strengthen the in silico analysis of the RPOTmp promoter with experimental validation to confirm the functionality of identified cis-regulatory elements in hormone responses​.
  • Explore further how the deficiency of RPOTmp alters the sensitivity of Arabidopsis to hormone treatments and its implications on overall hormonal balance and response mechanisms​.
  • Discuss the results from the GUS reporter strain analysis in the broader context of plant hormone signaling and mitochondrial gene expression, addressing potential limitations and alternative interpretations​.
  • Expand on the discussion regarding the independent regulation of nuclear and mitochondrial transcription, aligning it with existing literature and exploring potential mechanisms and implications​.
  • Improve overall clarity and coherence of the manuscript, integrating results with the introduction and discussion sections more fluidly and enhancing the accessibility of complex scientific content for readers​.

Author Response

We are sincerely grateful to the reviewer for carefully reading the manuscript and making critical comments and suggestions. We have made a number of corrections to the text and we hope to implement some of these very fair ideas in future studies.

  1. Clarify the rationale behind the selection of specific hormones and their concentrations used in the experiments, ensuring they reflect physiologically relevant conditions in Arabidopsis.

Response

The optimal concentrations of active reagents and duration of treatment were selected in preliminary experiments. The effectiveness of hormonal treatment was confirmed by expression analysis of standard marker genes ensuring physiologically relevant conditions for their application in Arabidopsis. The authors of the paper work in a laboratory that has been studying the mechanisms of action of phytohormones for over 50 years, and we certainly know that any experiment must be preceded by a concentration curve of phytohormone action.

  1. Provide detailed methodological explanations, especially regarding the qRT‒PCR analysis and the criteria for classifying genes as "regulated" based on transcript change ratios.

Response

The procedure of qRT‒PCR analysis is given in more detail in Materials and methods. Criteria for classifying genes as "regulated" was based on transcript change ratios. Differentially expressed genes with a ratio of transcript change of 1.5 in at least two tests were classified as regulated. A threshold of 1.5 is commonly used to identify differentially expressed genes (DEGs) when assessing transcriptomic data. For example, it was applied by Meredino et al (2020) to analyze transcriptomic data from rpoTmp mutant seedlings.

  1. Exercise caution in interpreting and generalizing results, particularly concerning the effect of hormone treatments on mitochondrial transcript abundance and the involvement of RPOTmp.

Response

All experiments were performed in three biological replicates and yielded similar results. The conclusions on the possible effect of hormone treatments on mitochondrial transcript abundance and the involvement of RPOTmp are drawn very carefully. We emphasized that the effect may represent the outcome of completely different effects, reflecting context dependent regulation of mt genes. We also concluded that the mutation of RPOTmp may induce a hormonal imbalance in concerted hormonal synthesis and signaling and, as a result, a differential mitochondrial response to hormonal treatment.

  1. Strengthen the in silico analysis of the RPOTmp promoter with experimental validation to confirm the functionality of identified cis-regulatory elements in hormone responses.

Response

Confirming the functionality of the identified cis-regulatory elements in hormonal responses requires additional time-consuming and labor-intensive experiments, such as one-hybrid assay, which are beyond the scope of this study and are planned for future work.

The objective of this work was to obtain primary information on the possible localization of active cis-elements of interest. Further work will include precise localization of cis-elements, production of new transgenic plants, and ultimately identification of hormone-dependent proteins that interact with the cis-elements under study. This is a very big job, it will be the subject of another article.

  1. Explore further how the deficiency of RPOTmp alters the sensitivity of Arabidopsis to hormone treatments and its implications on overall hormonal balance and response mechanisms.

Response

There was ample evidence of one hormone regulating genes involved in the metabolism of another hormone. This phenomenon has been suggested for many hormones and is particularly well-documented for the production of ethylene (Nemhauser et al, 2006)

While discussing the results of the experiments we suggested that  “The loss of RPOTmp has far-reaching implications for the activity of hormone-related genes. According to the data of Meredino et al. (2020), impaired function of RPOTmp caused changes reminiscent of the triple response of seedlings exposed to ethylene and could therefore contain increased levels of ethylene. In accord with extensive crosstalk and signal integration among growth-regulating hormones, plants with reduced or increased content of one hormone can show altered responses to another. Thus, rpotmp exhibited increased steady-state levels of transcripts for the CK marker gene ARR5, as well as elevated transcript accumulation of the CK synthesis genes IPT3 and IPT5 and a reduced level of hormone catabolism gene CKX3 expression, which implies a possible increase in the content of endogenous cytokinins [Bychkov et al,2021]. In parallel, the expression of reporter genes for GA (GA3) and SA (PR1) was changed in the rpotmp compared to WT (Figure 3). These results suggest that disruption of RPOTmp may induce a hormonal imbalance in concerted hormonal synthesis and signaling and, as a result, a differential mitochondrial response to hormonal treatment.”

A more detailed and substantiated analysis requires additional study of the content of hormones and of the expression of hormonal metabolism genes, which is the topic of a separate study

  1. Discuss the results from the GUS reporter strain analysis in the broader context of plant hormone signaling and mitochondrial gene expression, addressing potential limitations and alternative interpretations.

Response

Indeed, the GUS results should be interpreted with caution. We have added the following passage into the” Discussion”

It should be noted, as well, that lack of visible changes in GUS expression patterns may also be caused by a low promoter usage and does not necessarily correspond to the level of transcript accumulation. As stated by Lieber et al. (2018) “These two technologies focus on very different steps in gene expression (i.e., promoter usage versus transcript accumulation) and differences between them can be explained by, selective transcript stabilization or reduced transcript degradation”.

  1. Expand on the discussion regarding the independent regulation of nuclear and mitochondrial transcription, aligning it with existing literature and exploring potential mechanisms and implications.

Response

The discussion on the possible mechanisms of independent regulation of nuclear and mitochondrial transcription and its alignment with the existing literature is undoubtedly very promising and may represent the topic of an independent and very extensive review. However, it is not directly related to the detailed results presented in this study. Hence, interpretation of our data in the context of potential mechanisms and implications of nuclear and mitochondrial transcription will be extremely speculative and will contradict the principle “Exercise caution in interpreting and generalizing results”.

  1. Improve overall clarity and coherence of the manuscript, integrating results with the introduction and discussion sections more fluidly and enhancing the accessibility of complex scientific content for readers.

Response

To improve overall clarity and coherence of the manuscript we rearranged chapters 2.1 and partly 2.2 and also added the necessary explanations to the “Discussion” and” Materials and methods”.